# Molecular-Scale Investigations Reveal the Effect of Natural Polyphenols on BAX/Bcl-2 Interactions

**DOI:** 10.3390/ijms25052474

**Published:** 2024-02-20

**Authors:** Heng Sun, Fenghui Liao, Yichen Tian, Yongrong Lei, Yuna Fu, Jianhua Wang

**Affiliations:** Key Laboratory of Biorheological Science and Technology, Ministry of Education, College of Bioengineering, Chongqing University, Chongqing 400044, China; sunheng@cqu.edu.cn (H.S.); 20231901053g@stu.cqu.edu.cn (F.L.); 201919021135@cqu.edu.cn (Y.T.); 20161902085@cqu.edu.cn (Y.L.); fuyuna@cqu.edu.cn (Y.F.)

**Keywords:** atomic force microscopy, protein–protein interactions, BAX/Bcl-2, polyphenols

## Abstract

Apoptosis signaling controls the cell cycle through the protein–protein interactions (PPIs) of its major B-cell lymphoma 2-associated x protein (BAX) and B-cell lymphoma 2 protein (Bcl-2). Due to the antagonistic function of both proteins, apoptosis depends on a properly tuned balance of the kinetics of BAX and Bcl-2 activities. The utilization of natural polyphenols to regulate the binding process of PPIs is feasible. However, the mechanism of this modulation has not been studied in detail. Here, we utilized atomic force microscopy (AFM) to evaluate the effects of polyphenols (kaempferol, quercetin, dihydromyricetin, baicalin, curcumin, rutin, epigallocatechin gallate, and gossypol) on the BAX/Bcl-2 binding mechanism. We demonstrated at the molecular scale that polyphenols quantitatively affect the interaction forces, kinetics, thermodynamics, and structural properties of BAX/Bcl-2 complex formation. We observed that rutin, epigallocatechin gallate, and baicalin reduced the binding affinity of BAX/Bcl-2 by an order of magnitude. Combined with surface free energy and molecular docking, the results revealed that polyphenols are driven by multiple forces that affect the orientation freedom of PPIs, with hydrogen bonding, hydrophobic interactions, and van der Waals forces being the major contributors. Overall, our work provides valuable insights into how molecules tune PPIs to modulate their function.

## 1. Introduction

It is well known that the mitochondria-mediated apoptotic pathway plays an important role in programmed cell death. The B-cell lymphoma 2 protein family contains the pro-apoptotic protein BAX and the anti-apoptotic protein Bcl-2, which are major regulators in triggering the endogenous apoptotic pathway. During apoptosis, BAX proteins undergo conformational changes and migrate from the cytosol to the mitochondria, forming the mitochondrial pore [1]. Bcl-2 proteins act through heterodimerization with BAX to prevent pore formation, as well as the release of cytochrome c and the initiation of apoptosis [2,3]. The roles of BAX and Bcl-2 in cancer progression have been well documented [4,5], demonstrating their importance in apoptosis. Strategies to adjust the balance between BAX and Bcl-2 activities have been demonstrated to be associated with increased apoptosis susceptibility and drug-induced apoptosis resistance [6,7]. Therefore, the inhibition of the BAX/Bcl-2 interaction may restore the apoptotic instinct. 

The Bcl-2 family represents an ongoing area of research focused on regulating its protein–protein interactions. Over the past decades, researchers have focused on the inhibition of BAX/Bcl-2 complex formation by BH3 mimetics, as it has been shown to bind to the hydrophobic groove of Bcl-2 proteins, preventing BAX binding to Bcl-2 proteins [7,8,9]. Evidence collected so far indicates that PPI-targeted drugs are considered undruggable and do not follow Lipinski’s rule [10,11]. Only a few small molecule regulators have been developed for PPI-targeted therapy. The emergence of natural compounds has provided new options for drug development, especially some promising plant-derived compounds that trigger apoptosis while being non-toxic to healthy cells [12]. Gossypol is a pan-Bcl-2 inhibitor that blocks the BAX/Bcl-2 interaction [13]. Other studies have shown that polyphenols induce apoptosis in cancer cells by targeting Bcl-2 [4,13,14,15,16,17,18]. However, the molecular details of how polyphenols prevent the BAX/Bcl-2 interaction have not been clearly investigated, and little is known about the molecular mechanisms of polyphenol action and the quantitative interaction changes between paired proteins. These changes in the kinetics of protein intermolecular interactions are often accompanied by changes in non-covalent bonding (electrostatic, hydrophobic interactions, and hydrogen bonding). The modulation of PPI properties through drugs is regulated by complex energetics and usually involves changes in the macromolecular conformation in addition to the establishment of specific non-covalent bonds. Therefore, there is an urgent need to develop innovative platforms to elucidate the effects of drug binding on BAX/Bcl-2 interactions, as well as to induce conformational changes in proteins, which are essential for understanding its structure–function relationships and contribute to the design of effective drugs. 

In recent years, atomic force microscopy (AFM) has been proven to be a powerful approach for quantifying biomolecular bonds at the single-molecule level [19,20]. In the field of drug discovery, it also allows the evaluation of protein–drug interactions or drug-modulating PPIs under physiological or pathological conditions [21,22], providing valuable information on drug action. Previously, we reported the use of AFM to reveal the interaction force between BAX and Bcl-2 [23]. Through single-molecule force spectroscopy (SMFS), we revealed the existence of complex non-covalent interactions at the BAX/Bcl-2 interface. In this study, we aimed to derive the kinetic and thermodynamic parameters between BAX and Bcl-2 in the presence of different polyphenols (kaempferol, quercetin, dihydromyricetin, baicalin, curcumin, rutin, epigallocatechin gallate (EGCG), and gossypol) using force–distance (FD) curve-based AFM. We further employed contact angle molecular recognition and molecular docking to characterize the effect of polyphenols on the surface properties and interaction modes at the interface of the BAX/Bcl-2 complex. We found that polyphenols affect the orientation freedom of PPIs through multiple non−covalent forces, among which rutin, EGCG, and baicalin exhibited an excellent anti-binding ability to PPIs. This study provides theoretical and practical support for the drug discovery of PPIs. 

## 2. Results and Discussion 

### 2.1. Probing the Inhibition of BAX/Bcl-2 Binding by Polyphenols Using AFM 

The efficiency of polyphenols to inhibit BAX/Bcl-2 binding was investigated using the SMFS approach with the aim of understanding the single-molecule mechanisms by which small molecules modulate PPIs. A schematic of the AFM used to measure the inhibition of BAX/Bcl-2 by polyphenols is shown in Figure 1A. BAX and Bcl-2 were covalently functionalized to AFM tips and Au substrates, respectively, via N-hydroxysuccinimide surface chemistry. Combining the information obtained from the XPS and IR results (Appendix A), we confirmed that both proteins had been successfully immobilized on the Au substrate. 

During force measurement, the AFM tip first moves toward the protein on the substrate surface. The tip can then remain on the protein surface at a preset maximum contact force location for a short period of time (called contact time), providing sufficient time for protein–protein bonds to form. Subsequently, the tip retracts and disengages from the protein surface. The detector records the cantilever deflection during approach–retraction and generates FD curves. All raw FD curves were baseline−corrected using the JPK software (JPK, v7.0.97) by applying a linear drag to the last 30% of the retraction curve. Representative force curves are presented in Figure 1B, and the unbinding force can be derived from the jump-off in the force curves. FD curves that occur only in specific interaction pairs are categorized as specific binding events and are observed as a non-linear delayed retraction curve with a slope different from that of the contact region. We then identified specific FD curves as those that show discriminative patterns for specific binding pairs. Non−specific interaction force curves are generated by electrostatic interactions between the probe and substrate as previously described [24]. When BAX on the tip recognizes Bcl-2 on the substrate surface, the molecular recognition mapping appears as bright colored pixels. It can be seen that Bcl-2 is randomly distributed on the substrate surface (Figure 1C). As shown in Figure 1C, there are significantly fewer recognition events after polyphenol injection, and most force mapping pixels show no interactions. In evaluating the effect of the polyphenols on BAX/Bcl-2 interactions, all AFM measurements were performed in polyphenol physiological solutions, mimicking intracellular conditions. In the liquid phase, the probe was contaminated by polyphenol compounds, which can affect the force of the tip on proteins. This is difficult to avoid in single−molecule experiments. Therefore, we need to obtain a sufficient number of curves to exclude invalid data by distinguishing between force profiles characterized by the occurrence of interaction events. 

Binding probability (BP) is defined as the ratio between the number of unbinding events and the total of the recorded F-D curves. In 24.2% of the total curves, specific unbinding events of BAX/Bcl-2 interaction pairs are observed (Figure 1D). In our previous studies, we demonstrated the existence of a specific interaction between BAX and Bcl-2 using different system configurations (blocking and control experiments) [23]. After a gradually increase in the polyphenol injection concentration (ranging from 10 µM to 50 µM), we observed a progressive decrease in BP with polyphenol concentration (Figure 1D). As a positive control, for the specific inhibition capability of BAX/Bcl-2, we also tested a control gossypol, which has been reported to be a natural polyphenol that inhibits BAX/Bcl-2 interactions [4,25]. Using this control, we observed the specific inhibition, confirming that the inhibition of Bcl-2 was specific. A quantitative comparison of inhibition levels indicated that the BP of the BAX/Bcl-2 interaction was reduced by >50% at 50 μM, with the BP as follows: dihydromyricetin ≈ quercetin > kaempferol > curcumin > gossypol > baicalin > EGCG ≈ rutin, suggesting an IC_50_ in the μM range (Figure 1E). However, in the presence of quercetin and dihydromyricetin, BAX still binds to Bcl-2 with a 30–40% reduction in the BP. Similar results were also observed in previous publications on the inhibition of viral spike protein with angiotensin-converting enzyme-2 through antibodies [26,27,28]. Altogether, our in vitro assays at the single-molecule level provide direct evidence that polyphenols directly inhibit BAX/Bcl-2 binding, the protein binding site is shielded in the presence of polyphenols, and steric hindrance prevents BAX from reaching Bcl-2. 

### 2.2. Probing Single-Molecule Interaction Forces between BAX and Bcl-2 in Polyphenols

To quantify the binding interactions of BAX/Bcl-2 in the presence of polyphenols, differences in unbinding forces were measured for comparison. Typical FD curves are shown in Figure 2A, where the BAX/Bcl-2 pair interaction produced a specific force and sufficient data were collected for subsequent analysis. The average interaction force in the presence of polyphenols ranged from 60 to 80 pN, depending on the loading rate (LR) of 10 nN/s (Figure 2B). This is weaker than the force obtained on the BAX/Bcl-2 surface (82.4 pN). Single force curve measurements contain collective interaction forces from multiple pairs of BAX/Bcl-2 molecules, and thus, the average force values plotted in Figure 2B cannot be directly compared, where multiple bonds dissociation occur. Therefore, the data-fitting model (Gaussian mixture model) can be used to analyze the measurement data [29]. We assumed that the forces follow a mixed Gaussian distribution with multiple peaks corresponding to multivalent binding between protein pairs (Appendix A). Multivalent binding between BAX/Bcl-2 interfaces has been previously demonstrated [23]. We observed that the force distributions all exhibited periodic peaks. For the sake of brevity, we extracted the first peak force from the multi-Gaussian distribution, and the corresponding unbinding force of the single BAX/Bcl-2 complex was 71.35 ± 0.67 pN. While the peak forces for kaempferol, quercetin, dihydromyricetin, baicalin, curcumin, rutin, EGCG, and gossypol were 66.60 ± 0.64, 61.87 ± 4.50, 61.92 ± 1.71, 54.86 ± 1.29, 57.28 ± 1.19, 47.66 ± 0.86, 53.06 ± 0.57, and 58.59 ± 4.86 pN, respectively. These data are in the same strength as the interactions between amyloidogenic proteins [30] or redox proteins [31] and lower than the enzyme–coenzyme complexes [32], which is to be expected. Obviously, the addition of polyphenols resulted in decreases of 6.66%, 13.29%, 13.22%, 23.11%, 19.72%, 33.20%, 25.72%, and 17.88%, respectively. The unbinding forces remained similarly changed with different polyphenols, suggesting an enormous decreasing effect of exogenous molecules on the interacting forces of PPIs, and previous studies have yielded similar results [29,33].

In addition, the distribution of interaction force values is more dispersed due to the presence of complex interactions between BAX/Bcl-2, including both specific (hydrogen bonding) and non-specific (van der Waals, electrostatic interactions, and hydrophobic interactions) forces. In order to characterize the experimentally measured single-pair molecular forces, we assumed that the individual bonds of the BAX/Bcl-2 complex are independent of each other and that the number of bonds in the last unbinding event follows a Poisson distribution. We calculated the BAX/Bcl-2 single-molecular pair forces to gain insight into the effect of the polyphenols on the interaction (Figure 2C). The results revealed that the single-bond-specific (*F_i_*) and non-specific (*F*_0_) forces between BAX/Bcl-2 pairs in the control solution were 23.26 ± 0.58 pN and 67.15 ± 2.66 pN (Figure 2D). Specific forces in solutions of kaempferol, quercetin, dihydromyricetin, baicalin, curcumin, rutin, EGCG, and gossypol showed slight decreases to 20.72 ± 0.43, 18.94 ± 0.40, 22.62 ± 0.80, 18.22 ± 0.63, 19.76 ± 0.85, 14.99 ± 0.42, 16.29 ± 0.76, and 19.26 ± 0.20 pN, and the corresponding decrease rates were 10.92%, 18.56%, 2.76%, 21.68%, 15.05%, 35.57%, 29.96%, and 17.20%, respectively. Moreover, there were remarkable decreases in the non-specific forces in the polyphenols, 54.41 ± 1.86, 46.88 ± 1.71, 51.44 ± 2.99, 39.87 ± 2.44, 45.26 ± 3.39, 36.02 ± 2.11, 37.61 ± 3.32, and 49.94 ± 0.92 pN, resulting in decreases of 18.97%, 30.18%, 23.40%, 40.63%, 32.60%, 46.36%, 43.99%, and 25.63%, respectively, when compared to the control (Figure 2D,E). It can be inferred that polyphenols interfere more with non-specific interactions between BAX and Bcl-2 than with specific interactions, suggesting that hydrophobic interactions are the main drivers of PPIs. 

### 2.3. Probing the Kinetic Properties of Polyphenol-Regulated BAX/Bcl-2 Dissociation

Having demonstrated that polyphenols have the potential to hinder BAX/Bcl-2 interactions, we then investigated whether this difference originates from the dissociation and binding process of both proteins. Using DFS-based AFM, we measured the binding strength by imposing an external force on the BAX/Bcl-2 bonds and extracted the kinetic and energy landscape paraments of the interactions using the Friddle–Noy–de Yoreo (FNDY) model (Figure 3A), which describes the bond rebinding through two phases: an equilibrium phase at a lower LR, when the bond reversibly ruptures and rebinds, and a kinetic phase at a higher LR, when the bond irreversibly ruptures [34]. Experimentally, force curves were collected at different LRs (Figure 3B and Appendix A), and the most probable unbinding forces determined by the histogram Gaussian distribution were plotted against the LR, showing the force versus LR (Figure 3C–K). The BAX/Bcl-2 complex, among all polyphenols, can withstand forces of 30 to 150 pN in all polyphenols over the applied LR range, indicating a positive dependence of dynamic binding strength on LR, as previously observed for other systems [35,36]. Although forces in this range affect the conformational stability of the protein, the BAX/Bcl-2 binding interface remains mechanically stable, as previously demonstrated [23]. 

We fitted the data with the FNDY model and extracted the equilibrium forces ($F_{eq}$), dissociation rate constant ($k_{off}^{0}$), and barrier width ($x_{\beta }$) of the explored interaction pairs. The application of the FNDY model to the entire range of LR resulted in a high-quality fit to the DFS data. The introduction of polyphenols resulted in a reduction in $F_{eq}$, indicating that the force required to dissociate a single BAX/Bcl-2 bond to transition from equilibrium to a kinetic state is reduced. We hypothesized that multiple interactions of polyphenols in the final bound state would lead to a restriction of the orientation freedom between BAX and Bcl-2, and therefore change the kinetic of the BAX/Bcl-2 complex. To this end, we attempted to follow the dynamics of conformational changes based on the depth of the free energy valley of the stabilized complex and the free energy barrier properties that separate the bound state from its unbound states, determining the extent to which the BAX/Bcl-2 stable conformational state is populated. An increase in the $k_{off}^{0}$ was detected for the polyphenols in the following order: BAX/Bcl-2 ($k_{off}^{0}$ = 25.84 ± 1.90 s^−1^) < quercetin ≈ kaempferol ≈ dihydromyricetin < curcumin ≈ gossypol < baicalin < EGCG < rutin, with rutin and EGCG forming threefold fewer stable complexes. The single BAX/Bcl-2 bond lifetime (($\tau_{off}$ = ${1/k}_{off}$)) was accordingly decreased (Table 1). This indicates a high affinity and stability of the BAX/Bcl-2 complex with a longer lifetime for high-affinity interactions. The lower bond lifetime values indicate a tendency for rapid dissociation between BAX and Bcl-2 complexes in polyphenol solutions. Furthermore, a decrease in the $x_{\beta }$ was observed in the presence of polyphenols, indicating a change in the geometry/position of the binding pocket under these conditions. The narrower energy valley indicates that less conformational variability can be accommodated between the bound and transition states. As polyphenols hinder the interactions with BAX/Bcl-2, we observed a large free energy difference between bound and unbound state ($\Delta G_{bu}$), which reflects the number of amino acid residues associated with the binding interface of PPIs. Polyphenols reduce the number of amino acids with which BAX interacted with Bcl-2, increasing the $\Delta G_{bu}$ of BAX binding by a value of ~−16.65 Kcal/mol to −4~5 Kcal/mol. The higher energy barrier of BAX/Bcl-2 and the lower energy barrier in the presence of polyphenols are consistent with the binding site structure of the complex. BAX interacts with an extended hydrophobic groove on the Bcl-2 surface through its 20 Å-long α-helical-containing BH3 domains [37], which has a more well-defined binding site, resulting in a high affinity. Therefore, it is expected that the energy barrier of the BAX/Bcl-2 complex is higher. Variations in these parameters can be attributed to the restriction of the orientation freedom of BAX/Bcl-2 binding by polyphenol ligands.

### 2.4. Probing the Kinetic Properties of Polyphenol-Regulated BAX/Bcl-2 Association

Next, we intended to test whether our single-molecule approach was sufficiently sensitive to detect subtle differences in the association constant of the BAX/Bcl-2 complex when bound to different polyphenols, which provides a more accurate picture of the lifetime of the single bond. We observed that the BP increased exponentially with contact time (Figure 3), which can be described with Equation (9). Once τ was obtained from the exponential fit to the data, $k_{on}$ can be estimated by assuming that the interaction follows pseudo-first-order dynamics. It is clear from the shapes of the curves in Figure 3 that there are significant differences in the binding kinetics of BAX/Bcl-2 in the presence and absence of polyphenols. The results show that the binding in the control group reaches the plateau region earlier than in polyphenols, which indicates a substantially lower association rate of BAX/Bcl-2 in the polyphenol solutions. Simultaneously, these data strongly support our results on the dissociation rate constants as these confirm that the dissociation rate of BAX/Bcl-2 bonds is higher in polyphenol solutions. Collectively, these experiments contribute to the following equilibrium dissociation constants *K_D_* ($K_{D}=\frac{k_{off}}{k_{on}}$) in ascending order: BAX/Bcl-2 (~3.25 ± 0.4 × 10^−5^ M) < kaempferol < dihydromyricetin < quercetin < gossypol < curcumin < EGCG < rutin < baicalin (Table 2). This suggests that polyphenols disrupt but do not completely inhibit the binding of BAX to Bcl-2. Because Bcl-2 complexed with polyphenols is functionally inhibited, this suggests that the high-affinity binding of the natural ligand is required to completely inhibit Bcl-2. Altogether, our assays performed at the single-molecule level indicate that differences exist in the kinetics and energy landscape of BAX/Bcl-2 binding, with poorer binding capacities in the presence of different polyphenols compared to BAX/Bcl-2. 

### 2.5. Exploring the Polyphenols Modulating BAX/Bcl-2 Interfacial Properties

Having investigated the effect of polyphenols on the interaction forces within BAX/Bcl-2, we further investigated the effect of polyphenols on the surface tension of the BAX/Bcl-2 interface. The binding of polyphenol ligands may trigger changes in the microenvironment surrounding the protein. This approach has been previously used to analyze the inhibition of VEGF-A/VEGFR interactions by neutralizing anti-VEGF-A antibody [38]. As shown in Figure 4A, the water contact angle (θ_W_) indicated that the hydrophilic/hydrophobic properties of BAX/Bcl-2 were significantly changed in the presence of polyphenols. The θ_W_ of the untreated BAX/Bcl-2 surface was 67.46 ± 4.00, representing the wetting behavior of hydrophobic surfaces (θ_W_ > 65°) [39]. Polyphenol treatment decreases the hydrophobicity of the BAX/Bcl-2 surface, and this effect seems to be concentration dependent. Indeed, from the lowest polyphenol concentration (10 µM) to the highest (50 µM), the results showed a significant decrease in θ_W_ and increases in θ_EG_ and θ_D_ compared to the control group (Appendix A). 

The results for the surface tension component revealed that the Lifshitz–van der Waals component ($\gamma^{LW}$) for all the samples ranged in value from 35.9 to 40.4 mJ·m^−2^ (Figure 4B,C). The contribution of the Lifshitz–van der Waals force to the interfacial free energy is always attractive or zero [39]. In addition, the values of Lewis’s base (electron donor, $\gamma^{-}$) and Lewis’s acid (electron acceptor, $\gamma^{+}$) in the control group were 14.91 ± 0.30 and 0.16 ± 0.015 mJ/m^2^, respectively. $\gamma^{-}$ was significantly increased, and $\gamma^{+}$ was significantly decreased by the polyphenol treatment, and both the increase and decrease appeared to be concentration dependent (Figure 4D,E), which was attributed to the fact that the introduction of polyphenols increased the acid–base interaction functional groups. The increase in $\gamma^{-}$ confirms the weakening of the polar amino acid interaction of Bcl-2 with key residues of BAX. The PI values of BAX and Bcl-2 are less than 7, indicating that they are acidic in character [40]. A decrease in the acidic characteristics ($\gamma^{+}$) of the interface was observed after the introduction of polyphenols. Unlike the Lifshitz–van der Waals interaction, the contribution of acid–base interactions to the interfacial free energy can be attractive, repulsive, or zero [39]. Taking into account the standard deviations of the surface energy components ($\gamma^{LW}$, $\gamma^{-}$, and $\gamma^{+}$), the calculated changes of these components in the presence of polyphenols are evident. 

The affinity between two proteins can be described by their relative interaction free energy ($∆G_{i(W)i}$). When the relative $∆G_{i(W)i}$) value is negative, interactions are expected (i.e., potential spontaneous binding) [41]. When the ΔG value is positive, no affinity between proteins is expected. It is important to emphasize that the free energy of the interaction between BAX and Bcl-2 in water ΔG_Bcl-2(W)BAX_ are >0 at rutin, EGCG, and baicalin concentrations greater than 30 μM. (Figure 4C,F), indicating that they are hydrophilic. Qualitative analyses using θ_W_ values reveal that protein surfaces become hydrophilic after polyphenol treatment; in contrast, a ΔG_Bcl-2(W)BAX_ analysis demonstrated that the BAX/Bcl-2 interaction shift from hydrophobicity to hydrophilicity only in the presence of high polyphenol concentrations. This indicates that polyphenols occupy the hydrophobic groove of Bcl-2, whose binding pockets undergo significant deformation of the complex and rearrangement under tension. The decrease in hydrophobicity observed in this study may be attributed to the polyphenol binding to Bcl-2, which hinders the binding conformation of BAX to Bcl-2. The binding of polyphenols leads to an increase in free hydroxyl groups. These hydroxyl groups are hydrophilic due to their hydrogen bonding, thus reducing the hydrophobicity. 

### 2.6. Morphological Changes of Bcl-2 Protein

We performed tapping mode AFM imaging as a handy technique to evaluate the direct morphology of polyphenols binding with Bcl-2 at the single-molecule level without labeling and staining. Changes in shape and size are signs of polyphenol–protein binding interactions [22]. The obtained images illustrated that globular Bcl-2 proteins are identified as random distribution of small, sparse protrusions (Figure 5A), which is comparable to the AFM image simulated using BioAFMviewer software (v2.5) (Figure 5B). After incubation with polyphenol ligands, the shape of the Bcl-2 becomes swollen, suggesting a conformational change in the Bcl-2 complex (Figure 5C–J). Moreover, the increase in the number of bright spots indicates the formation of polyphenol–Bcl-2 complexes. The corresponding amplitude image and the supplemental 3D image (Appendix A) demonstrate similar regularities. 

Roughness (Sa, average roughness, and Sq, root mean square roughness) and average height were applied to quantify morphological changes in Bcl-2. Compared with free Bcl-2, Sa and Sq analyses of the spots revealed a corresponding increase in roughness by 1.2–2.8 nm after polyphenol injection (Figure 5K). The height of Bcl-2 is 10.82 ± 0.28 nm, while with the addition of polyphenols, the height of the complex varied from 11.69 to 14.74 nm (Figure 5L). The height of Bcl-2 macromolecules increased after interaction with the polyphenols, implying that the aggregation of Bcl-2 and polyphenols occurred during the interaction. This aggregation may arise from the tendency of Bcl-2 to form higher-order oligomers and protein–protein hydrophobic interactions. Similar results were obtained for the interactions of other active molecules with proteins, which clearly indicates that the molecular height of proteins is influenced by small molecules with affinity for the target protein [42,43]. Simultaneously, the interaction between polyphenols and protein complexes leads to a more hydrophobic microenvironment surrounding the proteins, as evidenced by the contact angle results. Bcl-2 may bind through hydrophobic interactions, and protein–water contact in the surface region is mostly minimized in the stable complex structure. It can be assumed that the molecular size of Bcl-2 becomes expanded when the polyphenols combine with Bcl-2, which is mainly due to hydrophobic and electrostatic forces. Similar phenomena were also observed in previous publications [44]. These results demonstrate the formation of complexes between polyphenols and proteins, suggesting that BAX/Bcl-2 interactions can be blocked through the formation of polyphenol/Bcl-2 complexes. 

### 2.7. Polyphenol Binding to Bcl-2 Competes for BAX

We used molecular docking to explore the structural mechanisms behind the polyphenol-regulated BAX/Bcl-2 interactions. It is well known that the classical hydrophobic pocket on the Bcl-2 monomer binds to the BH3 structural domain of BAX, and the major amino acid residues at this site include Arg107, Tyr108, Arg110, Asp140, Arg146, Leu201, and Tyr202 [45,46]. Small molecule compounds compete with the BH3 peptide in BAX to bind to the hydrophobic binding groove region of Bcl-2. Polyphenols (kaempferol, quercetin, dihydromyricetin, baicalin, curcumin, rutin, EGCG, and gossypol) were docked with Bcl-2 to explore the potential mode of interactions. The results revealed that the docking scores were 83.6, 82.7, 92.9, 113.2, 109.6, 129.7, 122.3, and 92.0, respectively. All eight polyphenols successfully binded to the outer surface cavity of Bcl-2 (Figure 6A–H). This corroborates with previous findings [13]. 

The binding kinetics between BAX/Bcl-2 are mainly mediated by hydrogen bonds, hydrophobic interactions, van der Waals forces, and electrostatic interactions [23]. Polyphenol binding to Bcl-2 alters the noncovalent interactions of PPIs, and multiple driving forces reduce the stability of the BAX/Bcl-2 complex. Rutin, curcumin, baicalin, and EGCG formed a greater number of amino acid interactions with Bcl-2 compared to the other compounds (Table 3). This result partially explains why these compounds have a stronger ability to inhibit BAX/Bcl-2. The molecular structures of kaempferol, quercetin and dihydromyricetin are highly similar. Based on the similarity in molecular structure and number of interacting amino acids, it is theorized that kaempferol, quercetin, and dihydromyricetin have similar affinities for Bcl-2. Overall, these molecular docking results are identical to the speculation and consistent with the interaction forces obtained via AFM, because these polyphenols form an important interaction network with the Bcl-2 interface. The binding of polyphenols affects the imaging and interaction force events of Bcl-2. The interaction force between BAX/Bcl-2 consists of multiple forces, with hydrogen bonding and hydrophobic interactions being the main ones. When polyphenols bind to Bcl-2, the formation of hydrogen bonds, van der Waals forces, and hydrophobic interactions weaken the interaction force between BAX/Bcl-2, leading to a decrease in the specific and non-specific force. These corresponding results explain the AFM results, revealing that the action of polyphenols is driven by hydrogen bonding, van der Waals forces, and hydrophobic interactions. In conclusion, our in vitro experiments at the single-molecule level provide direct evidence that polyphenolic compounds are potential candidates for inhibiting BAX binding to Bcl-2. 

## 3. Materials and Methods

### 3.1. Materials and Reagents

16-Mercaptohexadecanoic acid (MHA), N-Hydroxysuccinimide (NHS), 1-Ethyl-3-(3-dimethylaminopropyl) carbodiimide (EDC), ethanolamine, ethylene glycol, and diiodomethane were purchased from Sigma-Aldrich (Saint Louis, MO, USA). Polyphenols (kaempferol, quercetin, dihydromyricetin, baicalin, curcumin, rutin, EGCG, and gossypol) were acquired from Shanghai Yuanye Bio-Technology Co., Ltd. (Shanghai, China). Stock solutions of 5 mM were prepared by dissolving polyphenols in dimethyl sulfoxide (DMSO), and then the working solutions of polyphenols were diluted in phosphate buffer (pH 7.2–7.4, 0.01 M) and kept in the dark at 4 °C. BAX and Bcl-2 human recombinant proteins were purchased from Proteintech (Wuhan, China). The 15 µg/mL solutions of BAX and Bcl-2 were prepared in phosphate buffer (pH 7.2–7.4, 0.01 M). AFM probes (ContGB-G, Tap150Al-G and HQ: CSC38) were purchased from BudgetSensors (Sofia, Bulgaria) and MikroMasch (Watsonville, CA, USA). Gold-coated substrates were purchased from Ted Pella, Inc. (16010-G, Redding, CA, USA). 

### 3.2. Immobilization of Proteins on AFM Tips and Substrate

AFM tips and substrates were prepared as previously described [23,47]. Briefly, Au-coated tips (ContGB-G with tip radius < 25 nm, an apex half-cone angle of 10° and resonance frequency of 13 kHz, and HQ: CSC38 with tip radius < 35 nm, full cone angle of 40° and resonance frequency of 10 kHz) and substrates were rinsed with acetone, ethanol, piranha solution (H_2_SO_4_:H_2_O_2_ = 7:3, *v/v*), and ultrapure water to remove contaminants. Subsequently, a self-assembled monolayer was formed via incubation with 1 mM MHA for 24 h and then washed several times with ethanol/water to remove free MHA. A mixture of NHS (10 mg/mL) and EDC (25 mg/mL) was added and incubated for 30 min at room temperature to activate terminal carboxyl groups and couple -NH_2_ groups exposed on proteins. Finally, the treated tips and substrates were inserted into protein solution (1 µM) and incubated overnight at 4 °C. The remaining activated carboxyl groups were deactivated with 1 M ethanolamine. Followed by washing with PBS, the functionalized tips and substrates were used for force spectroscopy measurements or stored in PBS buffer at 4 °C until use. 

### 3.3. Single-Molecule Force Spectroscopy Measurements between BAX an Bcl-2 on Model Substrates

All unbinding force measurements were executed using a JPK NanoWizard 2 (Berlin, Germany) equipped with a piezoelectric scanning head with a maximum z-range of 15 μm. The effective spring constant (*k_eff_*) of the functionalized probes (ContGB-G and HQ: CSC38) was determined using thermal noise measurement. The LR was the product of the *k_eff_* and the retraction velocity (*v*). To estimate the kinetic parameters, unbinding forces were measured under the same approaching velocity (2 µm/s) and different retraction velocity (ranged from 0.1 to 10 μm/s), with a threshold force of 0.3 nN, and tip contact time remained at 500 ms. Additionally, unbinding forces were measured at different contact times (varying from 0 to 1 s) to estimate association rate constant, while the retraction velocity was set at 1 μm/s. The BAX/Bcl-2 competitive binding assay was performed in a polyphenol (30 µM) PBS solution containing 1% Triton X-100 at room temperature. Force-mapping mode images were taken on a 32 × 32 grid over a 500 × 500 nm^2^ scanning area, obtaining 1024 force curves over the region. All samples were run at least three times. All force curves were analyzed using JPKSPM Data Processing software (JPK, v7.0.97). 

### 3.4. Morphological Changes in Protein

The tapping model of AFM was employed to clarify the topographic changes in Bcl-2 protein after polyphenol binding. In total, 2 μL of protein and polyphenol (30 µM) were mixed for 30 min, deposited on freshly cleaved mica, and imaged with NanoWizard 2 AFM (JPK, Berlin, Germany) in tapping mode at room temperature. 

A rectangular Si_3_N_4_ cantilever with a spring constant of 4 N/m (Tap150Al-G, with a height of 17 μm, a nominal radius of <10 nm, an apex half-cone angle of 10° and resonance frequency of 150 kHz) was chosen for protein imaging. The samples were scanned using a line frequency of 1 Hz with a resolution of 512 × 512 pixels. All images were analyzed using the Gwiddion software v2.58. BioAFMviewer [48] software (v2.5) was used to generate simulated AFM images of Bcl-2 proteins. 

### 3.5. Theoretical Model Used to Quantitatively Analyze the Interactions

To characterize the experimentally measured force distributions, we assumed that the individual bonds of the tip and the substrate are independent of each other, that the number of bonds in the unbinding event follows a Poisson distribution [29,49,50], and that interactions involving multiple bonds can be represented as follows: (1)$$P\left(N\right)=\frac{e^{-\lambda }\lambda^{N}}{N!}$$
(2)$$\sigma_{n}^{2}=\lambda$$
where $P\left(N\right)$ is the probability of forming N bonds, and λ and $\sigma_{n}^{2}$ represent the mean and variance of the bonds formed in the interaction events, respectively. Then, the unbinding force values will follow the following equations: (3)$$F_{av}=\lambda F_{i}+F_{0}$$
(4)$$\sigma_{F}^{2}={\left(\sigma_{n}F_{i}\right)}^{2}=\lambda F_{i}^{2}$$
(5)$$\sigma_{F}^{2}=\lambda F_{i}^{2}=F_{av}F_{i}-F_{i}F_{0}$$
where $F_{i}$ is the single-bond specific force, $F_{0}$ is the non-specific force, $F_{av}$ is the mean value of the force, and $\sigma_{F}^{2}$ is the variance of the force. 

The Friddle–Noy–de Yoreo (FNDY) model [34] describes the force-induced reversible bond rupture and rebinding events. The bond rupture process is divided into two phases: an equilibrium phase at a low LR, in which the bond is broken and rebound, and a kinetic phase at a high LR, in which the bond is irreversibly ruptured. The transition between the two phases occurs at the equilibrium force ($F_{eq}$) and is described as follows: (6)$$F_{eq}=\sqrt{2k_{eff}\Delta G_{bu}}$$
where $k_{eff}$ is the effective spring constant for the whole system, and $\Delta G_{bu}$ is the equilibrium free energy between the unbound and bound states. Under the equilibrium force ($F_{eq}$), the molecular complex transits from the near-equilibrium state to the kinetic state: (7)$$F≅F_{eq}+f_{\beta }\ln\left(1+e^{-\gamma }\frac{r}{k_{off}\left(F_{eq}\right)f_{\beta }}\right)$$
where $f_{\beta }$ is the thermal force scale, $f_{\beta }=\frac{k_{B}T}{x_{\beta }}$, $k_{off}\left(F_{eq}\right)$ is the force-induced dissociation rate, and γ = 0.577 is Euler’s constant. The dissociation rate ($k_{off}$) in the absence of an external force is given as follows:(8)$$k_{off}=\frac{k_{off}\left(F_{eq}\right)}{\exp\left[\frac{1}{k_{B}T}\left(F_{eq}x_{\beta }-\frac{k_{eff}{x_{\beta }}^{2}}{2}\right)\right]}$$

For association rate analysis, the relationship between the interaction time and BP is described by the following equation [26]: (9)$$BP=A*\left[1-\exp\left(\frac{-\left(t-t_{0}\right)}{\tau }\right)\right]$$
where A is the maximal BP, $t_{0}$ is the lag time, and τ is the time required for the half-maximal binding probability. The value of $k_{on}$ is estimated by applying the following expression: (10)$$k_{on}=\frac{V_{eff}\cdot N_{A}}{n_{b}\tau }$$
where $V_{eff}$ is the effective volume of a sphere describing the protein binding pair, $N_{A}$ is Avogadro’s number, and $n_{b}$ is the number of binding partners. 

### 3.6. Interfacial Property Measurements Using Contact Angle

The Bcl-2-functionalized substrates were incubated with polyphenols for 30 min and then incubated with BAX proteins for 30 min. The substrates were washed with buffer and blown dry with nitrogen, and then, contact angle measurements were performed immediately. The static contact angles under the three liquids (water, ethylene glycol, and diiodomethane) were measured at room temperature (25 °C) using a contact angle meter equipped with a CCD camera (SDC-200S, Shengding Precision Instruments Co., Ltd., Dongguan, China). Three probe liquids were utilized to probe the different components of surface energy. The surface tension components of these liquids can be obtained from the literature [41]. The components of the total surface energy were determined from Van Oss, and the Young equation used for the calculations was as follows [51,52,53,54]:(11)$$0.5\left(1+cos\theta \right)\gamma_{L}=\sqrt{\gamma_{S}^{LW}\gamma_{L}^{LW}}+\sqrt{\gamma_{S}^{+}\gamma_{L}^{-}}+\sqrt{\gamma_{S}^{-}\gamma_{L}^{+}}$$
where θ is the contact angle between the droplet (L) and the surface of the component (S), γ is the surface tension, LW is the Lifshitz–Van der Waals component, and + and − are the electron acceptor and electron donor components, respectively. The interfacial free energy of interaction between two molecules in water is determined according to the following equation [38,41,55]: (12)$$∆G_{i(W)i}=2(\sqrt{\gamma_{1}^{LW}\gamma_{w}^{LW}}+\sqrt{\gamma_{2}^{LW}\gamma_{w}^{LW}}-\sqrt{\gamma_{1}^{LW}\gamma_{2}^{LW}}-\gamma_{w}^{LW}+\sqrt{\gamma_{w}^{+}}\left(\sqrt{\gamma_{1}^{-}}+\sqrt{\gamma_{2}^{-}}-\sqrt{\gamma_{w}^{-}}\right)\;+\sqrt{\gamma_{w}^{-}}\left(\sqrt{\gamma_{1}^{+}}+\sqrt{\gamma_{2}^{+}}-\sqrt{\gamma_{w}^{+}}\right)-\sqrt{\gamma_{1}^{+}\gamma_{2}^{-}}-\sqrt{\gamma_{2}^{+}\gamma_{1}^{-}})$$

### 3.7. Molecular Docking

Molecular docking was used to investigate the binding of polyphenol ligands with the Bcl-2 using Discovery Studio 2019. The 3D structures of the polyphenols (kaempferol, quercetin, dihydromyricetin, baicalin, curcumin, rutin, EGCG, and gossypol) were obtained from PubChem, and the molecule structures were optimized. The X-ray crystal structure of Bcl-2 (PDB ID: 6O0K) was downloaded from the Protein Data Bank. The protein structure was pre-processed prior to docking to remove the water and proto-ligand and to add all hydrogen atoms. Binding sites were determined from the literature [23,46], and the active sites were defined as a sphere of 15 Å. The docking parameters were set as follows: 100 hotspots, 0.25 Å docking tolerance, high-quality docking preference, and FAST conformation generation method, with the number of conformations set to 255. Other parameters were set to their default values. The conformation with the highest LibDockScore was deemed the best conformation. 

### 3.8. Data Analysis

Force data were fitted to DFS models using the nonlinear curve fitting module of OriginPro 2019b (OriginLab, Nothampton, MA, USA). Statistical significance was calculated by performing unpaired *t*-tests using GraphPad Prism version 8.3.0 (GraphPad Software, La Jolla, CA, USA). 

## 4. Conclusions

The multiple functions of Bcl-2 family proteins that contribute to this pathway are excellent therapeutic targets, and the selective inhibition of the Bcl-2 family proteins through small molecules is a promising new approach in drug discovery. However, there is limited information on how molecules interact with their targets at the molecular scale. In our study, we developed a single-molecule approach to analyze the perturbation of PPIs through small molecules using AFM. Several important physical parameters related to the kinetics and thermodynamics of the binding and dissociation of BAX/Bcl-2 in polyphenol solutions were quantified. The kinetics of the binary complexes are consistent with a nonlinear single-barrier two-state transition model, and by comparing the thermodynamic and kinetic parameters measured under different scenarios, we confirm that the polyphenol is a competitive inhibitor of Bcl-2, preventing the further binding of BAX once bound. The mechanistic insights gained directly confirm quantitative views on how small molecules regulate PPIs to populate the free energy landscape in different ways. Our experiments arrive at the consistent conclusion that the interference of polyphenols with PPIs is driven by a combination of hydrophobic interactions and hydrogen bonding and restricts the orientation freedom of BAX/Bcl-2. The order of inhibition of BAX/Bcl-2 by polyphenols is as follows: rutin > EGCG > baicalin > curcumin > gossypol > quercetin ≈ dihydromyricetin ≈ kaempferol. The differences in the inhibition of PPIs are due to differences in the way the polyphenols interact with Bcl-2, masking or altering the hydrophobic cavity region of the Bcl-2 structure, which is ultimately attributed to structural differences in the polyphenols studied. 

Given the differences in the AFM experiments performed on different polyphenols, we believe that the scientific value of AFM lies not in obtaining the absolute biophysical parameters of PPIs system, but rather in using it as a toolbox for comparing how variables affect these characteristics. For the design of drug-targeted PPIs, it is important to utilize quantitative biophysical parameters measured at the single-molecule level to assess whether a compound has the desired drug-like molecular properties. Our approach may complement the traditional rules for the small-molecule modulation of PPIs, which provides an interesting insight into the use of AFM as a screening tool. However, the regulation of intracellular protein interactions is complex, and its structural and functional states depend on the biological environment, i.e., cells, organelles, and cell membranes. Indeed, exploring a range of information is required to understand it. Therefore, we may not have measured the interaction forces of small molecules on the complete formation and migration of BAX/Bcl-2 in the natural cellular context. This also suggests that further integrative studies are needed in the future to measure biophysical parameters across spatial scales (single protein, mitochondrial, cellular structure) via AFM, and to utilize this knowledge to formulate strategies for the small-molecule regulation of the life cycle. 

## Figures and Tables

**Figure 1 ijms-25-02474-f001:**
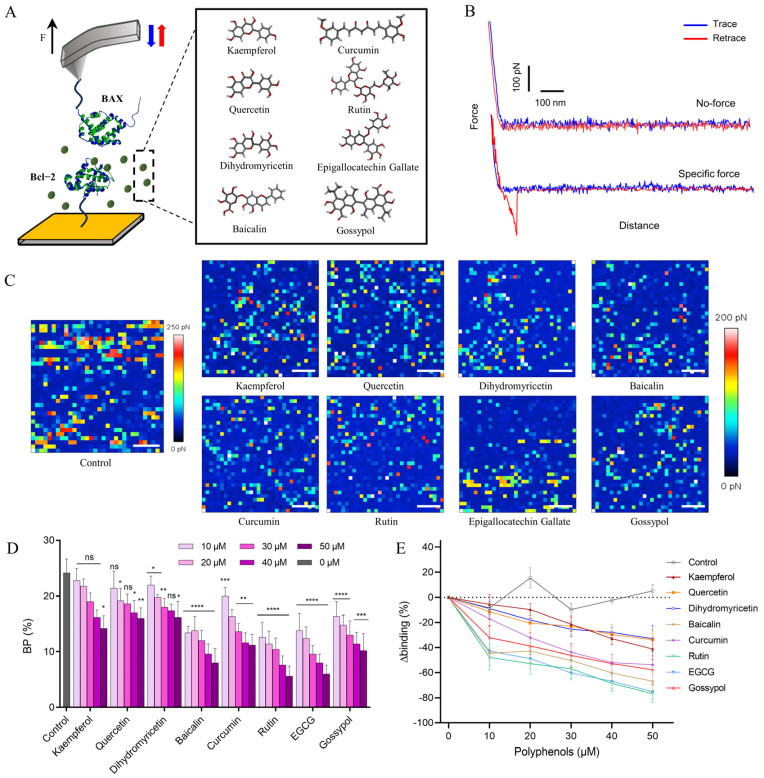
Probing the efficiency of polyphenols in inhibiting BAX binding to Bcl-2. (**A**) Schematic of probing BAX binding to Bcl-2 in the presence of polyphenols using AFM. (**B**) Representative F−D curves illustrating non−specific and specific events. (**C**) Interaction force mapping of BAX/Bcl-2 before and after polyphenol treatment. Scale bars represent 500 nm. (**D**) BP histogram of BAX/Bcl-2 interaction in different polyphenol solutions. (ns: not statistically significant, * indicates *p*-values < 0.05; ** indicates *p*-values < 0.01; *** indicates *p*-values < 0.001, **** indicates *p*-values < 0.0001 on unpaired sample *t*-tests). (**E**) Graph showing the reduction in the binding frequency. Data are representative of at least N = 3 independent experiments per polyphenol concentration. The error bars indicate the s.d. of the mean value.

**Figure 2 ijms-25-02474-f002:**
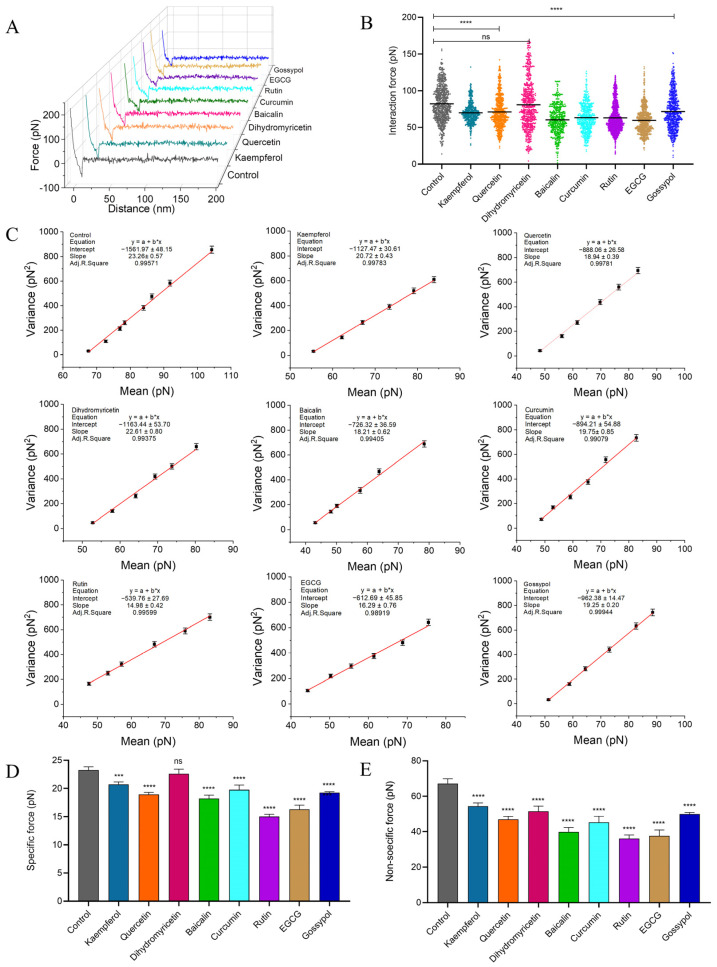
Unbinding force measurements of the BAX/Bcl-2 in the presence of different polyphenols. (**A**) Representative F−D curves before and after polyphenol treatment. (**B**) Measured unbinding forces for all individual curves collected at a LR of 10 nN/s; the horizontal lines represent the means. (**C**) Linear relationship of the mean (λ) plotted versus the variance ($\sigma_{F}^{2}$) of the unbinding forces in the control or polyphenols. (**D**,**E**) Bars represent the specific (**D**) and non−specific (**E**) forces between BAX/Bcl-2 pairs in the control and polyphenols (ns: not statistically significant, *** indicates *p*-values < 0.001, **** indicates *p*-values < 0.0001 on unpaired sample *t*-tests).

**Figure 3 ijms-25-02474-f003:**
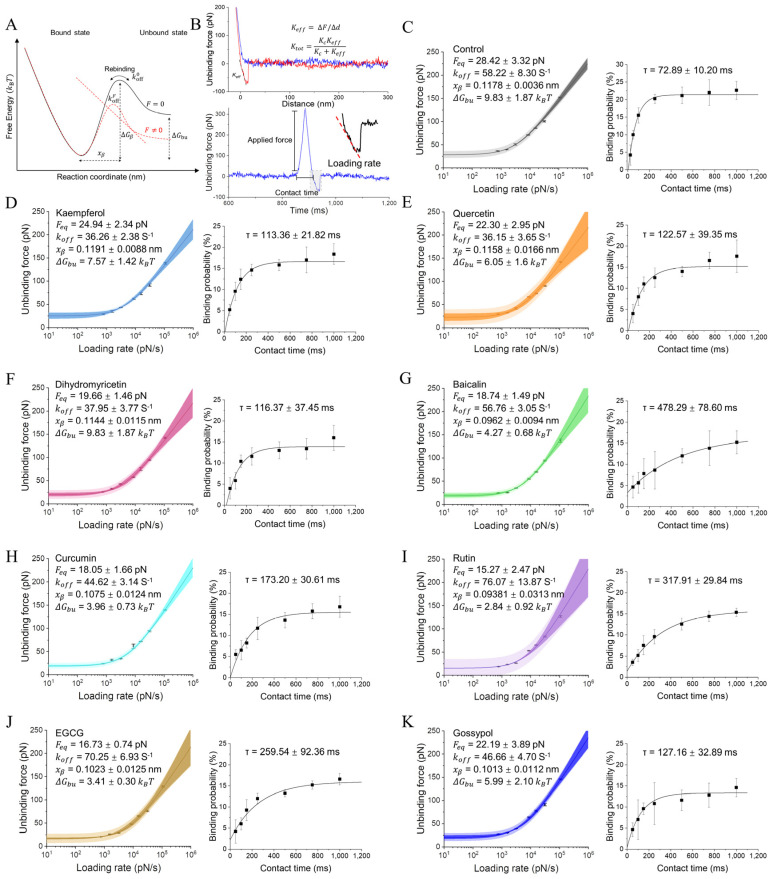
Quantification of the subtle kinetic differences of BAX/Bcl-2 interactions under different polyphenols. (**A**) The binding of BAX to Bcl-2 can be described using a simple two-state model. The bound state is in an energy valley with an energy barrier between it and the unbound state. (**B**) $K_{eff}$ is extracted from the FD curve and used to infer the spring constant $K_{tot}$ for the whole system. The FD curve can be displayed as a force–time curve, from which the LR can be extracted by bonding the slope of the curve prior to rupture. (**C**–**K**) Dynamic force spectroscopy plot showing the LR-dependent interaction forces of BAX/Bcl-2 bonds at seven distinct LRs in the presence of polyphenols, control (**C**), kaempferol (**D**), quercetin (**E**), dihydromyricetin (**F**), baicalin (**G**), curcumin (**H**), rutin (**I**), EGCG (**J**), and gossypol (**K**). Data corresponding to single interactions were fitted with the FNDY model, providing $F_{eq}$, $k_{off}$, $x_{\beta }$, and $\Delta G_{bu}$ values. Darker shaded areas represent 95% confidence intervals, and light shaded areas represent 95% of the prediction intervals of the fit. Plots on the right: BP as a function of contact time. Least-squares fitting of the data to a mono-exponential decay curve yield τ. All experiments were repeated at least three times using independent tips and samples. Error bars indicate the mean s.d.

**Figure 4 ijms-25-02474-f004:**
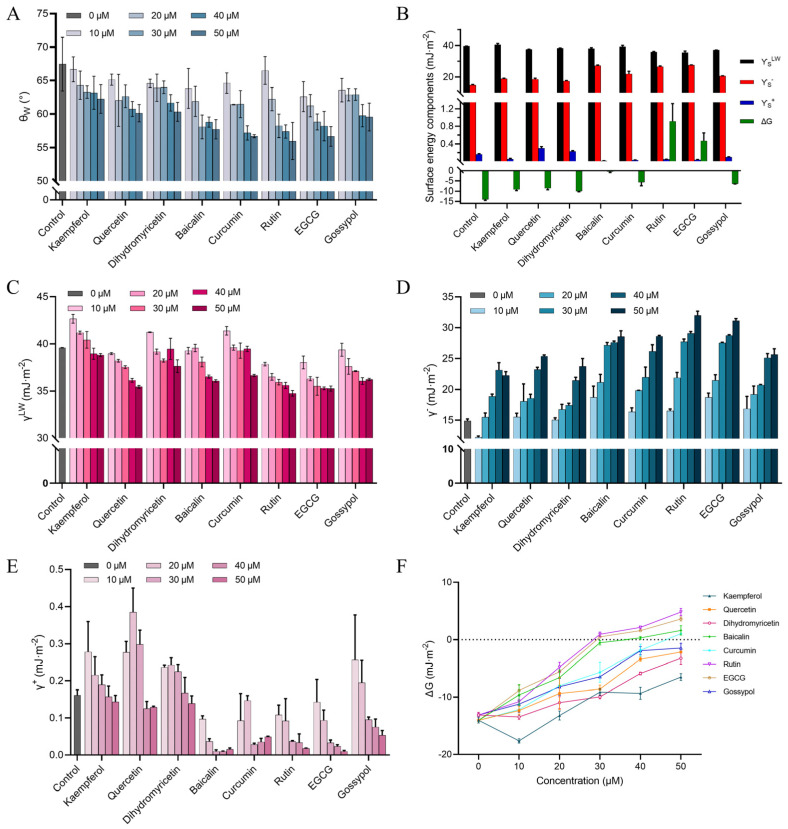
Quantification of the surface parameters and interaction free energy of BAX/Bcl-2 interactions under different polyphenols. (**A**) Static contact angles of water measured on BAX/Bcl-2 surfaces before and after polyphenol treatment. (**B**) Interaction surface free energies and their components calculated from the Van Oss theory, (at 30 µM polyphenols). (**C**–**F**) Lifshitz–van der Waals values (**C**), electron donor values (**D**), electron acceptor values (**E**), and interaction free energy values (**F**) at different polyphenol concentrations.

**Figure 5 ijms-25-02474-f005:**
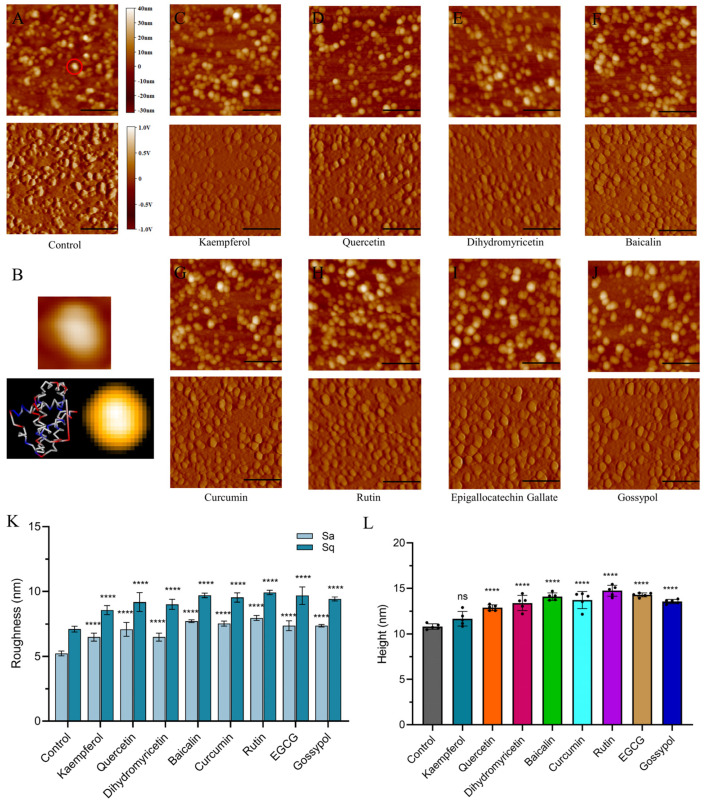
AFM imaging reveals morphological changes in Bcl−2 protein. (**A**) Free Bcl−2. (**B**) Enlarged view of the morphology of Bcl−2 in Figure 5A (red−circled area). Crystal structure of the Bcl−2 (PDB ID: 2XA0) conformation was shown in the 3D structures, colored by charge, and the structure was used to simulate an AFM image (simulation parameters were a scanning step of 1 nm, cone angle of 20°, and a tip radius of 10 nm). (**C**–**J**) Imaging of Bcl−2 in different polyphenol solutions (30 µM): Bcl−2 with baicalin, EGCG, curcumin, kaempferol, quercetin, rutin, dihydromyricetin, and gossypol, respectively, and corresponding amplitude images (bottom). Scale bars represent 500 nm. (**K**) Histogram of surface roughness parameters of Bcl−2 in different polyphenol solutions. (**L**) Histogram of average height of Bcl−2 in different polyphenol solutions. (ns: not statistically significant, **** indicates *p*-values < 0.0001 on unpaired sample *t*-tests).

**Figure 6 ijms-25-02474-f006:**
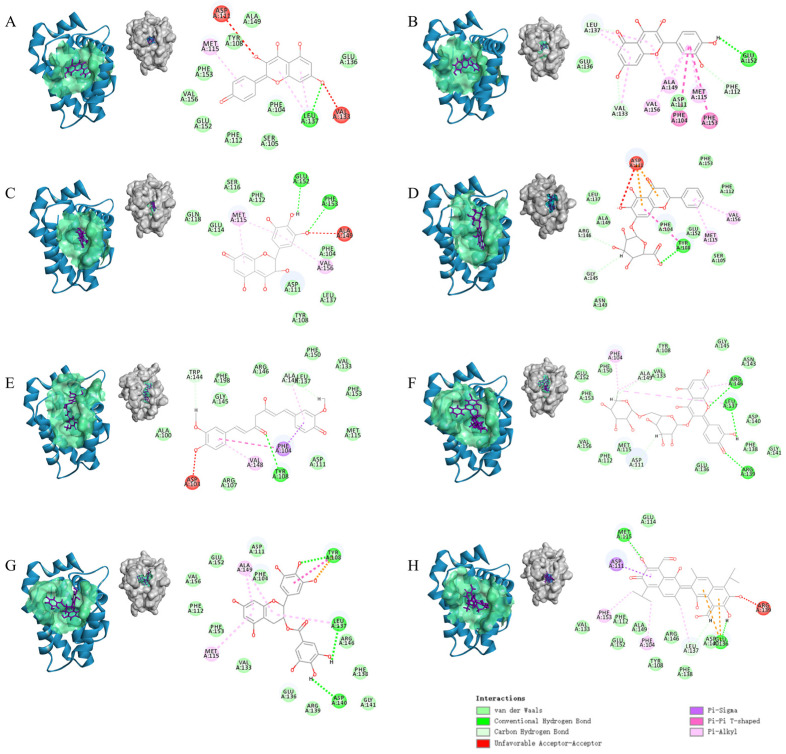
The lowest energy docking complex conformations of polyphenols binding to Bcl-2. (**A**–**H**) are the complexes of Bcl-2 with kaempferol (**A**), quercetin (**B**), dihydromyricetin (**C**), baicalin (**D**), curcumin (**E**), rutin (**F**), EGCG (**G**), and gossypol (**H**), respectively (left side). In the corresponding 2D schematic (right side), light green circles correspond to amino acid residues that interact with polyphenols via van der Waals forces. Green dashed lines indicate hydrogen bonds, and purple and orange dashed lines indicate π-π stacking interactions and π-cation interactions, respectively.

**Table 1 ijms-25-02474-t001:** Comparison of kinetic and energy landscape paraments of BAX/Bcl-2 interactions under different polyphenol conditions.

Sample	$F_{eq}$ (pN)	$f_{\beta }$ (pN)	*k_off_* (*F_eq_*) (S^−1^)	$k_{off}$ (S^−1^)	$x_{\beta }$ (nm)	$\tau_{off}$ (S)	−Δ*G*_bu_ (*k*_B_*T*)
Control	28.42 ± 3.32	34.25 ± 1.87	58.22 ± 8.30	25.84 ± 1.90	0.1178 ± 0.0036	0.0387	9.83 ± 1.87
Kaempferol	24.94 ± 2.34	34.49 ± 2.62	73.44 ± 13.56	36.26 ± 2.38	0.1191 ± 0.0088	0.0276	7.57 ± 1.42
Quercetin	22.30 ± 2.95	35.49 ± 4.95	66.66 ± 18.31	36.15 ± 3.65	0.1158 ± 0.0166	0.0277	6.05 ± 1.60
Dihydromyricetin	19.66 ± 1.46	35.93 ± 3.65	64.54 ± 12.36	37.95 ± 3.77	0.1144 ± 0.0115	0.0264	4.70 ± 0.70
Baicalin	18.74 ± 1.49	42.71 ± 4.19	87.03 ± 11.22	56.76 ± 3.05	0.0962 ± 0.0094	0.0176	4.27 ± 0.68
Curcumin	18.05 ± 1.66	38.24 ± 4.32	70.53 ± 11.51	44.62 ± 3.14	0.1075 ± 0.0124	0.0224	3.96 ± 0.73
Rutin	15.27 ± 2.47	43.81 ± 16.25	106.64 ± 36.90	76.07 ± 13.87	0.09381 ± 0.0313	0.0131	2.84 ± 0.92
EGCG	16.73 ± 0.74	40.18 ± 4.98	105.19 ± 17.33	70.25 ± 6.93	0.1023 ± 0.0125	0.0142	3.41 ± 0.30
Gossypol	22.19 ± 3.89	40.56 ± 4.53	79.63 ± 20.15	46.66 ± 4.70	0.1013 ± 0.0112	0.0214	5.99 ± 2.10

**Table 2 ijms-25-02474-t002:** Comparison of association rate and dissociation constants for characterizing the interactions of BAX/Bcl-2 under different polyphenol conditions.

Sample	τ (ms)	$k_{on}$ (M^−1^s^−1^)	$K_{D}$ (M)
Control	72.89 ± 10.20	7.96 ± 0.93 × 10^5^	3.25 ± 0.44 × 10^−5^
Kaempferol	113.36 ± 21.82	5.12 ± 0.84 × 10^5^	7.09 ± 1.25 × 10^−5^
Quercetin	122.57 ± 39.35	4.73 ± 1.41 × 10^5^	7.64 ± 2.88 × 10^−5^
Dihydromyricetin	116.37 ± 37.45	4.98 ± 1.49 × 10^5^	7.61 ± 2.39 × 10^−5^
Baicalin	478.29 ± 78.60	1.21 ± 0.17 × 10^5^	4.68 ± 0.70 × 10^−4^
Curcumin	173.20 ± 30.61	3.35 ± 0.50 × 10^5^	1.33 ± 0.20 × 10^−4^
Rutin	317.91 ± 29.84	1.82 ± 0.14 × 10^5^	4.17 ± 0.83 × 10^−4^
EGCG	259.54 ± 92.36	2.23 ± 0.75 × 10^5^	3.14 ± 0.74 × 10^−4^
Gossypol	127.16 ± 32.89	4.56 ± 1.04 × 10^5^	1.02 ± 0.38 × 10^−4^

**Table 3 ijms-25-02474-t003:** The binding patterns of polyphenols to Bcl-2.

Polyphenols	The Number of Interacting Amino Acids	Amino Acids Forming H-Bond	Amino Acids Involved in van der Waals Interactions	Amino Acids Involved in Hydrophobic Interactions	Binding Energy (kcal/mol)
Kaempferol	11	Leu137	Ala149, Tyr108, Phe153, Val156, Glu152, Phe112, Ser105, Phe104, Glu136	Met115, Leu137	−26.47
Quercetin	11	Phe112, Val133, Leu137, Glu152	Leu137, Glu136, Asp111, Val133, Phe112	Phe104, Met115, Val133, Leu137, Ala149, Phe153, Val156	−27.72
Dihydromyricetin	12	Glu152, Phe153	Gln118, Ser116, Glu114, Phe112, Phe104, Asp111, Leu137, Tyr108	Met115, Val156	−27.50
Baicalin	13	Tyr108, Gly145, Arg146	Leu137, Ala149, Asn143, Phe104, Glu152, Ser105, Phe112, Phe153	Tyr108, Met115, Val156	−41.32
Curcumin	16	Tyr108, Trp144, Ala149	Ala100, Arg107, Phe198, Gly145, Arg146, Asp111, Leu137, Phe150, Val133, Phe153, Met115	Phe104, Val148, Ala149	−33.66
Rutin	21	Asp111, Leu137, Arg139, Arg146, Ala149	Glu152, Phe150, Phe153, Val156, Phe112, Met115, Tyr108, Val133, Gly145, Asn143, Asp140, Phe138, Gly141, Glu136	Phe104, Leu137, Arg146, Ala149	−64.85
EGCG	17	Tyr108, Leu137, Asp140,	Glu152, Val156, Phe112, Phe153, Val133, Asp111, Phe104, Glu136, ARG139, Gly141, Phe138, Arr146	Tyr108, Met115, Leu137, Ala149	−56.17
Gossypol	14	Met115, Glu136, Leu137	Glu114, Val153, Phe112, Glu152, Ala149, Arg146, Tyr108, Phe138, Asp 140	Phe104, Asp111, Met115, Leu137, Phe153	−34.37

## Data Availability

The original contributions presented in the study are included in the article/Appendix A; further inquiries can be directed to the corresponding author.

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
