# Peer review of "Molecular-Scale Investigations Reveal the Effect of Natural Polyphenols on BAX/Bcl-2 Interactions"

_ijms, 2024, doi:10.3390/ijms25052474_

Round 1

Reviewer 1 Report

Comments and Suggestions for Authors

The manuscript titled “Molecular-scale Investigations Reveal the Effect of the Natural Polyphenols on BAX/Bcl-2 Interactions” by Sun, H.; et al. is a scientific work where the authors study the impact of the presence of polyphenols with different chemistry nature (kaempferol, quercetin, dihydromyricetin, baicalin, curcumin, rutin, epigallocatechin gallate and gossypol) on the interaction established between the B-cell lymphoma 2 pro-apoptotic (BAX) and anti-apoptotic (Bcl-2) proteins which is a key factor to have a more complete outlook about the underlying mechanisms related to human cancer diseases.This is a topic of growing interest and this work could be interesting for a certain target audience and gain impact in the near future.

However, it exists some points that need to be addressed (please, see them below detailed point-by-point) to improve the scientifc quality of the submitted manuscript paper before this article will be consider for its publication in the International Journal of Molecular Sciences.

1) ABSTRACT. “Apoptosis (…) BAX and Bcl-2” (lines 8-9). The full-name of these proteins should be stated as in the Introduction section (line 29). Then, the abbreviations should be placed between brackets.

2) INTRODUCTION. This section clearly depicts the state-of-the-art of this field with relevant citation references. No actions are requested from the authors.

3) RESULTS AND DISCUSSION. “During force measurement (…) generate force-distance curves. Representative force curves are presented in Fig. 1B and the unbinding force can be derived from the jump-off in the force curves” (lines 88-94). All the force curve profiles collected by the authors were similar than the Fig. 1B? Was neccesary to conduct a further baseline correction prior the force-curve analysis in any specific condition case? A brief statement should be provided in this regard.

4) Figure 1 (line 122). The lateral scale bar should be indicated in the force-volume maps indicated in panel C. Similar comment in the Figure S6 (Supplmentary information) where the vertical height should be displayed for all the tested conditions.

5) “The average interaction (…) loading rate of 1 µm/s” (lines 136-138). Please, the authors should not confuse the retraction velocity (in µm/s) respecting to the loading rate (typically in nN/s). The authors should take into account the spring constant of the used cantilever after its proper calibration by the thermal tune (as indicated in the line 443) to exchange among these parameters.

6) “We observe that the force distribution (…) unbinding force of single BAX/Bcl-2 complex was 71.35 ± 0.67 pN. While the peak force of kaempferol (…) 58.59 ± 4.86 pN, respectively” (lines 146-151). Here, a brief discussion with the unbinding forces reported to other biomolecular complexes will benefit to the potential readers to better understand the strength of the specific interactions measured by the authors about BAX/Bcl-2 complex and in presence of different polyphenols. In this context, the interactions gathered by the authors exhibit the same strength as the exerted between amyloidogenic proteins [1] or redox proteins [2] and lower than enzyme:coenzyme systems [3] which is as expected.

[1] Doherty, C.P.A.; et al. A peptide-display protein scaffold to facilitate single molecule force studies of aggregation-prone peptides. Protein Sci. 2018, 27, 1205-1217. https://doi.org/10.1002/pro.3386.

[2] Bonanni, B.; et al. Single molecule recognition between cytochrome C 551 and gold-immobilized azurin by force spectroscopy. Biophys. J. 2005, 89, 2783-2791. https://doi.org/10.1529/biophysj.105.064097.

[3] Pérez-Domínguez, S.; et al. Nanomechanical Study of Enzyme:Coenzyme Complexes: Bipartite Sites in Plastidic Ferrodoxin NADP+ Reductase for the Interaction with NADP. Antioxidants 2022, 11, 537. https://doi.org/10.3390/antiox11030537.

7) Figure 2, panel C (line 178). The standard deviation (SD) bars should be added in each condition of all the measured samples. Same comment for the plots according to the binding probability vs the dwell-time in Figure 3 (line 236).

8) Then, did the authors observe a contamination of the polyphenols present in the liquid media (e.g. by the direct immobilization at the functionalized tips by electrostatically interactions) during the acquisition of the force-spectroscopy data which could negatively affect to the data interpretation? Some discussion should be furnished about this regard.

9) Then, the authors could provide the regression coefficient (R2) related to all the Gaussian distributions shown in Fig. S3? This could positively aid to see the quality of all the force fittings.

10) Table 2 (line 277). Please, the authors should add superscripts in the data related to the association rate and the dissociation constant.

11) MATERIALS AND METHODS. “AFM tips (…) Au-coated tips (…)” (lines 429-431). The model of the AFM probes used in the force spectroscopy and AFM imaging (lines 455-463) measurements should be indicated.

12) “3.7. Molecular docking” (lines 508-516). How many number of interactions were counted to carry out the docking simulation experiments?

13) CONCLUSIONS. This section clearly outlines the most relevant outcomes found in this work. The authors should discuss about some potential future action lines to pursue this research. Finally, the citation references are in the proper format of the International Journal of Molecular Sciences (No actions are requested from the authors).

Comments on the Quality of English Language

The manuscript is generally well-written albeit it may be advisable if the authors could recheck it in order to polish final details susceptible to be improved.

Reviewer 2 Report

Comments and Suggestions for Authors

The manuscript examines the use of natural polyphenols (kaempferol, quercetin, dihydromyricetin, baicalin, curcumin, rutin, epigallocatechin gallate, and gossypol) to regulate the binding process between BAX and Bcl-2. The authors utilized AFM as a technique to assess how polyphenols affect the BAX/Bcl-2 binding mechanism. Additionally, the authors developed a single-molecule approach to analyze how small molecules perturb the BAX/Bcl-2 interaction.     

     The kinetics studies of the binary complexes provide evidence that polyphenols act as competitive inhibitors of Bcl-2, preventing BAX from binding once it has already bound.

     The present study indicates that the interference of polyphenols with PPIs involves both hydrophobic interactions and hydrogen bonding, which restrict the freedom of orientation for BAX/Bcl-2. The polyphenols' inhibition of BAX/Bcl-2 follows this order: rutin > EGCG > baicalin > curcumin > gossypol > quercetin ≈ dihydromyricetin ≈ kaempferol. The variations in inhibition of PPIs are a result of the different ways in which the polyphenols interact with Bcl-2, either masking or altering the hydrophobic cavity region of Bcl-2's structure. Ultimately, these differences can be attributed to the structural variances among the polyphenols studied.

     Using AFM as technique for this study is relevant and interesting. The paper is well written, and the text is clear and easy to read. The conclusions are consistent with the evidence and arguments presented.

     I agree with the publication of this manuscript in International Journal of Molecular Science after some minor modifications, as follows:

1.      Please improve the resolution of all figures.

2.      The meaning of BAX and Bcl-2 abbreviations should be made in Abstract and not in Introduction.

Round 2

Reviewer 1 Report

Comments and Suggestions for Authors

The authors did a great deal of effort to cover all the raised suggestions made by the Revieweres. Subsequently, the scientific quality of this manuscript was greatly improved. For all the above described reasons and based on the significance of this research in the examined field, I would like to warmly endorse this work for further publication in the International Journal of Molecular Sciences.